# Biofilm Production and Its Implications in Pediatrics

**DOI:** 10.3390/microorganisms12081522

**Published:** 2024-07-25

**Authors:** Nicola Principi, Susanna Esposito

**Affiliations:** 1Università degli Studi di Milano, 20122 Milan, Italy; nicola.principi@unimi.it; 2Pediatric Clinic, Department of Medicine and Surgery, University of Parma, 43126 Parma, Italy

**Keywords:** biofilm, acute otitis media, adenoiditis, cystic fibrosis, otitis media with effusion, protracted bacterial bronchitis

## Abstract

Biofilms, aggregates of bacteria enclosed in a self-produced matrix, have been implicated in various pediatric respiratory infections, including acute otitis media (AOM), otitis media with effusion (OME), adenoiditis, protracted bacterial bronchitis, and pulmonary exacerbations in cystic fibrosis. These infections are prevalent in children and often associated with biofilm-producing pathogens, leading to recurrent and chronic conditions. Biofilms reduce antibiotic efficacy, contributing to treatment failure and disease persistence. This narrative review discusses biofilm production by respiratory pathogens such as *Streptococcus pneumoniae*, non-typeable *Haemophilus influenzae*, *Pseudomonas aeruginosa*, and *Staphylococcus aureus*. It examines their mechanisms of biofilm formation, antibiotic resistance, and the challenges they present in clinical treatment. Various antibiofilm strategies have shown promise in vitro and in animal studies, including the use of N-acetylcysteine, enzymes like dispersin B, and agents disrupting quorum sensing and biofilm matrix components. However, their clinical application, particularly in children, remains limited. Traditional treatments for biofilm-associated diseases have not significantly evolved, even with biofilm detection. The transition from experimental findings to clinical practice is complex and requires robust clinical trials and standardized biofilm detection protocols. Addressing biofilms in pediatric respiratory infections is crucial for improving treatment outcomes and managing recurrent and chronic diseases effectively.

## 1. Introduction

Since the late seventeenth century, it has been demonstrated that several microorganisms are capable of forming biofilms—aggregates of one or more bacterial species attached to a surface and enclosed in a self-produced matrix [1]. Subsequent studies have further characterized the mechanisms through which microbial cells attach to living tissues and medical devices, develop a community structure and ecosystem, and eventually detach from the initial biofilm colony [2].

The biofilm matrix acts as a barrier, isolating microorganisms from their environment, and making them more resistant than planktonic bacteria to antibiotics, host defenses, and external stressors such as pH, osmolarity, nutrient availability, and mechanical shear forces [3]. Although these findings have primarily been obtained from in vitro studies or experiments on animals [4], they are considered robust enough to suggest that biofilms can cause significant clinical problems in humans.

Biofilms are thought to reduce the clinical efficacy of antibiotics and promote the development of recurrent or chronic infections. This is supported by evidence that biofilms are more commonly detected in subjects with difficult-to-treat infections compared to those without. Numerous studies have been conducted to identify which pathogens can form biofilms, understand how this process occurs, and determine which measures might prevent, inhibit, and eradicate biofilms to reduce the risk of chronic or recurrent infections [5].

In pediatric cases, particular attention has been given to respiratory infections, as these are the most common bacterial diseases in children and are frequently recurrent or chronic [6,7,8]. This narrative review discusses biofilm production by bacteria that commonly cause respiratory infections in children. It also examines the antibiotic resistance of these pathogens and their role in causing difficult-to-treat respiratory infections in children. Finally, the review explores current antibiofilm measures used in pediatric cases. A comprehensive literature search was conducted using the MEDLINE (Northfield, IL, USA) and PubMed (Bethesda, MD, USA) databases, covering the period from 1993 to 30 April 2024. The search included randomized placebo-controlled trials, controlled clinical trials, double-blind, randomized controlled studies, systematic reviews, and meta-analyses. Abstracts published in the last two years were included. The following combinations of keywords were used: “biofilm” AND “*Pseudomonas aeruginosa*” OR “*Staphylococcus aureus*” OR “otitis media” OR “protracted bronchitis” OR “cystic fibrosis”.

## 2. Biofilm Formation by Respiratory Bacteria

In otherwise healthy children, the primary bacteria responsible for respiratory infections are *Streptococcus pneumoniae* (*Sp*), non-typable *Haemophilus influenzae* (nt*Hi*), and *Streptococcus pyogenes* (*Spy*). *Mycoplasma pneumoniae* and *Moraxella catarrhalis* (*Mc*) also contribute but play a minor role [9]. In patients with conditions such as bronchopulmonary dysplasia, bronchiectasis, or cystic fibrosis (CF), the most common causes of respiratory infections are *Pseudomonas aeruginosa* (*Pa*) and *Staphylococcus aureus* (*Sa*) [10]. All these pathogens are capable of forming biofilms through different genetically programmed mechanisms (Table 1).

The biofilm produced by respiratory bacteria is a complex, structured community of microbial cells enclosed within a self-produced polymeric matrix [11,12]. This matrix, composed of polysaccharides, proteins, and extracellular DNA, adheres to both biotic and abiotic surfaces in the respiratory tract. It provides a protective environment, enhancing bacterial survival and resistance to antibiotics and immune responses. Biofilm formation on respiratory tissues generally follows the same steps. It begins with the attachment of planktonic bacteria to respiratory cells. Initially, attachment is via the cell pole or the flagellum and is reversible. Later, attachment becomes longitudinal and irreversible, primarily due to reduced flagella reversal rates, decreased flagella gene expression [11], and the role of bacterial adhesins [12]. As flagella gene expression diminishes, bacterial cells start producing increasing amounts of matrix components that enable them to adhere to each other and form a multilayer biofilm. This stage, known as maturation, involves cell-to-cell communication through a quorum-sensing system. This system relies on diffusible signal molecules that regulate the activation of genes required for matrix production, motility, and virulence factors. The biofilm matrix consists of extracellular polymeric substances (EPS), including exopolysaccharides, nucleic acids (eDNA and eRNA), proteins, lipids, and other biomolecules, with significant variations between bacteria [13]. Over time, the biofilm degrades spontaneously due to reduced nutrient resources and the accumulation of toxic substances. This leads to the dispersal of bacterial cells from the biofilm, allowing them to spread and potentially cause new infections or form new biofilms [14,15].

Different pathogens exhibit distinct genetically programmed mechanisms for biofilm formation and evolution. For instance, in the upper respiratory tract, *Streptococcus pneumoniae* biofilm formation depends on DNA adenine methyltransferase, which regulates the biosynthesis of the quorum-sensing molecules Com and LuxS/AI-2. These molecules play a fundamental role in biofilm development. Biofilm formation in nt*Hi* is critically dependent on the type IV pilus, an adhesin that facilitates pathogen adhesion and biofilm formation [16]. Additionally, the DNABII family of nucleoid-associated proteins is essential for maintaining biofilm structure [17]. The stability of *Pa* biofilms is determined by various polysaccharides, including alginate, pel, and psl. Alginate is crucial for preserving the matrix’s structure and contents, while pel and psl serve as primary structural scaffolds. Extracellular DNA is also vital for cell-to-cell interconnection and biofilm maturation [18]. Biofilm formation in *Sa* depends on several regulatory factors, including the accessory quorum sensing system. Intercellular adhesion requires the polysaccharide intercellular adhesin (PIA), produced and regulated by the intercellular adhesion (ica) ADCB operon. Surface components like *Staphylococcal* protein A (spa) also contribute to adhesion within biofilms [19].

Each pathogen utilizes specific mechanisms for biofilm formation and maintenance, highlighting the complexity and diversity of bacterial biofilm development in respiratory infections.

## 3. Increased Resistance to Antibiotics in Respiratory Bacteria Living in Biofilms

Bacteria living in biofilms are generally 10 to 1000 times more resistant to antibiotics, host immune defenses, and adverse environmental conditions (e.g., high temperature, low pH, high salinity, pressure, and nutrient scarcity) than their planktonic counterparts [20]. In vitro studies have identified several factors responsible for this biofilm-related tolerance or resistance to antibiotics, including poor penetration of antimicrobials and immune defenses through the biofilm matrix, differential physiological activity of bacteria within biofilms, differential gene expression, and the development of persister cells, which are bacteria in a dormant state [21].

Understanding the distinction between tolerance and resistance is crucial when bacteria in biofilms are dispersed into the environment. Regardless of whether they are tolerant or resistant, bacteria within biofilms exhibit reduced sensitivity to one or more antibiotics. However, tolerant bacteria, when dispersed, often gradually regain their previous sensitivity. If they cause a new infection, it can typically be treated with recommended doses of commonly prescribed antibiotics, with no greater risk of treatment failure than infections caused by planktonic bacteria. In contrast, bacteria that have developed resistance remain resistant and can cause severe, difficult-to-treat infections. This contributes to the increasing prevalence of antibiotic-resistant pathogens and highlights why biofilm formation is a significant factor in the rise of antimicrobial resistance [22,23].

The biofilm matrix’s poor permeability can significantly reduce antibiotic concentrations within the biofilm. This reduction may occur when antibiotics bind to matrix components or are inactivated by enzymes in the matrix, such as beta-lactamases [24]. For example, amoxicillin, commonly prescribed for bacterial respiratory infections in infants and children, can have its efficacy reduced due to poor matrix penetration. However, poor penetration is not the primary cause of tolerance, as many antibiotics, including fluoroquinolones and rifampin, can penetrate the matrix well but still fail to eradicate all bacteria within the biofilm [25,26].

Within biofilms, oxygen and nutrient levels are typically low, and metabolic byproducts and waste products accumulate. These conditions negatively affect antibiotics that target processes in actively growing bacteria, such as replication and cell wall synthesis. Additionally, low pH and oxygen tension can impair antibiotic efficacy. For instance, tobramycin and ciprofloxacin are less effective in anaerobic environments, and aminoglycosides are less effective under conditions of low pH and oxygen tension [27,28,29]. In vivo, the tolerance observed in vitro can be significantly greater due to additional factors affecting bacterial metabolism and antibiotic diffusion through the matrix.

In conditions such as cystic fibrosis (CF), where *Pa* infections are common, immune responses can create anaerobic conditions at infection sites. Polymorphonuclear leukocytes (PMNs) consume oxygen, creating local anaerobic conditions that affect bacterial metabolism and biofilm growth. Low oxygen levels have been observed in the maxillary sinuses of CF patients with chronic rhinosinusitis, wound tissue, and infected bones [30,31]. Low oxygen tension also inhibits the production of highly deleterious hydroxyl radicals by Gram-negative and Gram-positive bacteria, reducing the efficacy of bactericidal antibiotics [32]. Furthermore, low oxygen concentrations impede the transport of certain antibiotics, like aminoglycosides, across bacterial membranes [33].

Persister cells are bacterial cells without metabolic activity, making them tolerant to drugs targeting actively growing bacteria. Bacteria can become persisters spontaneously or in response to antibacterial agents in biofilms. To survive, bacteria activate intracellular toxin proteins that arrest cell growth and induce a dormant state [34]. Various genera, including *Escherichia coli*, *Pa*, *Sa*, *Lactobacillus acidophilus*, and *Gardnerella vaginalis*, can form persister cells [35].

Resistance development can result from spontaneous mutations or horizontal gene transfer (HGT). Mutations may occur when bacteria in biofilms are exposed to subtherapeutic antibiotic levels, as seen when antibiotics at recommended doses poorly penetrate the matrix. Additionally, bacterial responses to DNA damage from negative environmental factors can induce significant structural and functional gene modifications. These modifications help to maintain DNA integrity but increase mutagenesis, leading to altered antibiotic targets, enhanced efflux pump production, and changes in bacterial cell wall characteristics [36].

HGT is more common in biofilms than in planktonic cultures, facilitated by the polymicrobial nature of biofilms, increased bacterial density, and the presence of environmental DNA in the matrix. Key mechanisms of HGT include conjugation (direct gene transfer between cells), transformation (acquisition of DNA from the environment), and transduction (gene transfer via bacteriophages). Membrane vesicles and nanotubes also play roles in HGT, acting as DNA reservoirs and facilitating direct cell-to-cell contact, respectively [37].

These mechanisms collectively contribute to the increased antibiotic resistance seen in biofilm-associated bacteria, posing significant challenges in treating biofilm-related infections.

## 4. Measures to Inhibit Biofilm Formation and Disrupt Preformed Biofilm

In recent years, our understanding of the mechanisms by which bacteria form biofilms has significantly increased, leading to the development of numerous strategies to inhibit biofilm formation or disrupt preformed biofilms [38]. Most studies on this topic have been conducted in vitro or in experimental animals, with very few examples in humans.

Measures aimed at inhibiting biofilm formation generally interfere with one or more stages of the biofilm life cycle. An exception is N-acetylcysteine, which has been found capable of both inhibiting biofilm formation and disrupting preformed biofilms [39]. Strategies to limit biofilm development include inhibiting bacterial surface adhesion, interfering with the quorum-sensing system, remodeling cellular metabolism, and disrupting biofilm maturation. Commonly studied measures include artificial or natural surfactants, compounds targeting host-EPS interactions, bacterial adhesin inhibitors, small-molecule inhibitors of EPS glucan synthesis, and quorum-sensing system disruptors. Comprehensive analyses of these approaches have been published by Koo et al. [40] and Ghosh et al. [41].

In clinical practice, these measures are considered potentially effective for preventing biofilm formation after surgery or on medical devices. However, they are less suitable for established mature biofilm conditions, such as those seen in children with recurrent or chronic respiratory diseases. In these cases, strategies targeting the disruption of mature biofilms are more appropriate. This can be achieved through physical removal, EPS-matrix degradation, biofilm dispersion induction, and killing persister cells.

Physical removal, such as scraping or brushing, aims to dismantle the basic biofilm structure, facilitating its chemical removal. While too invasive for living tissues, this method is effective for biofilm disruption on medical devices or teeth [42]. Enzymes have been suggested for degrading the EPS-matrix, based on the observation that several enzymes are naturally secreted by bacterial cells within biofilms and are essential for EPS-matrix remodeling. Engineered enzymes like dispersin B, deoxyribonuclease I (DNase I—dornase alfa), and alginate lyase have shown effectiveness in disrupting biofilms, though with varying efficacy across different bacteria. Dispersin B significantly degrades *Staphylococcus epidermidis* biofilms by specifically disrupting poly-N-acetylglucosamine (PNAG), the main exopolysaccharide of this bacterium [43]. However, it is less effective against *Burkholderia cenocepacia* and *Achromobacter xylosoxidans* biofilms, as PNAG is not a major component of these pathogens [44].

The activity of enzymes can be significantly enhanced when combined with other compounds that are effective against biofilm-producing bacteria or that facilitate enzyme adhesion or penetration into the matrix. For instance, combining dispersin B with a silver-binding peptide that promotes the in situ formation of silver nanoparticles (AgNPs) in the presence of silver ions resulted in at least a twofold increase in activity against 48-hour-old *S. epidermidis* biofilms compared to the enzyme alone [45]. Similarly, combining DNase I or alginate lyase with vancomycin decreased the matrix material and significantly reduced the minimal inhibitory concentrations for *Enterococcus faecalis* and *Enterococcus faecium* [46]. Additionally, combining alginate lyase with ciprofloxacin effectively reduced the biomass and cell density of biofilm-grown *Pa* due to improved antibiotic penetration into the biofilm [47].

The disruption of biofilms can also be achieved by targeting bacterial intracellular signaling pathways that facilitate the transition from planktonic to biofilm states [48]. Reducing bis-(3-5)-cyclic dimeric GMP (c-di-GMP) is a potential strategy. Both azathioprine and nitric oxide (NO) have been found effective in this regard. Azathioprine decreases bacterial c-di-GMP, correlating with the reduced expression of extracellular matrix proteins in *Escherichia coli* [49]. NO donor prodrugs, direct NO inhalation, and increasing endogenous NO formation are potentially effective alternatives, as several in vitro and animal studies have shown NO can promote biofilm dispersal [50].

Other approaches to biofilm destruction include using compounds that disrupt quorum sensing, such as garlic extract and certain antibiotics [51]. Azithromycin, for example, although not effective against *Pa*, can interfere with quorum sensing and block alginate production in experimental infections caused by this pathogen [52].

Additionally, reducing iron availability can destroy biofilms. Studies have shown that iron depletion in models of *Pa* and *Burkholderia cenocepacia* infections promotes the transition from biofilm to planktonic states. Gallium, which mimics iron properties, can be taken up by bacteria instead of iron, creating an iron-deficient population. In an in vitro study using lung epithelial cells, adding gallium to gentamicin was more effective than the antibiotic alone in eradicating Pseudomonas aeruginosa biofilms.

Finally, bacteriophages can eradicate biofilms through multiple mechanisms, targeting bacterial cells and encoding various enzymes, such as depolymerases and lysins, which recognize, bind, and disturb biofilm structures [53].

## 5. Pediatric Respiratory Diseases Associated with Mucosal Biofilm Detection

Biofilms have been repeatedly detected on the respiratory mucosa of children with recurrent or chronic respiratory infections, although the incidence rate of biofilm detection is significantly influenced by the methods used [54]. Currently, there is no standardized protocol for diagnosing biofilms, and results from studies using different techniques can vary widely, making comparisons difficult and results debatable. Additionally, some proposed methodologies are expensive, time-consuming, and require a large number of trained personnel, making them impractical for clinical practice [55]. Further studies are needed to identify a protocol that ensures rapid biofilm detection and increases the chances of successful infection control. Among pediatric biofilm-associated diseases, acute otitis media (AOM), otitis media with effusion (OME), chronic adenoiditis, protracted bacterial bronchitis, and cystic fibrosis are the most common.

### 5.1. Recurrent Acute and Chronic Suppurative Otitis Media

Acute otitis media (AOM) is the most common bacterial infection among infants and children. By the age of three, nearly all children will have experienced at least one episode of AOM, and about 50% will suffer from recurrent episodes (rAOM) [56]. In some cases, persistent ear discharge through a perforated eardrum can last for 2–6 weeks or more, leading to chronic suppurative otitis media (csOM). Approximately 70% of AOM cases are caused by bacteria, mainly *Sp*, nt*Hi*, *Mc*, and *Spy*. Pa and Sa play a minor role, although they are frequently detected in csOM [56]. The role of biofilms in rAOM and csOM has been supported by several in vitro and animal studies [57].

One of the strongest demonstrations that biofilms may influence the development and progression of middle ear diseases came from evidence of mucosal biofilms in the nasopharynx (58%) and middle ear (47%) of chinchillas with induced middle ear inflammation and/or infection [58]. Human studies have repeatedly confirmed these findings. In a study involving children with and without rAOM, biofilms were significantly more common in the nasopharynx of those with recurrences compared to healthy controls (41.4% vs. 14.5%; *p* < 0.005), with nt*Hi* (50.0%) and *Spy* (37.5%) being the most common biofilm-producing bacteria [59]. Chronic adenoiditis or adenoid hypertrophy can significantly contribute to biofilm-related rAOM and csOM (see the following section), although middle ear biofilm-associated diseases can occur regardless of adenoid involvement. Another study in children with normal adenoid size showed that the prevalence of biofilm-producing otopathogens in the nasopharynx was higher in the rAOM group than in controls (29.3% vs. 10.9%; *p* = 0.02) [59].

While the importance of biofilm-producing otopathogens in causing severe and recurrent ear problems has been definitively established, several aspects of the pathogenic role of biofilms remain unresolved [60]. For example, it is not yet clear whether the degree of biofilm formation or merely the capability of producing a biofilm by common respiratory pathogens is relevant in influencing AOM treatment failure or rAOM [61,62]. Moreover, since biofilms are only detected in a portion of children with rAOM and csOM, it is uncertain whether this is due to the inefficacy of detection methods or because biofilms are not essential for the development of chronic or recurrent ear diseases. Resolving these issues could significantly enhance the treatment approach for these conditions and the use of recently developed antibiofilm measures.

Currently, studies on the use of these antibiofilm preparations in children with biofilm-related middle ear problems are limited. Considering that biofilm matrices contain bacterial or host-derived DNA, it has been hypothesized that using a recombinant DNase, such as Dornase alfa, could effectively degrade biofilms and significantly improve standard antibiotic therapy. However, despite promising in vitro studies, the results of administering a single aural dose of Dornase alfa in children undergoing ventilation tube insertion for rAOM or csOM were not entirely satisfactory. Hearing improved in all children following ventilation tube insertion, with no indication of ototoxicity. Dornase alfa had some effect on increasing the time until VT extrusion (*p* = 0.099) and blockage and/or extrusion (*p* = 0.122), but the frequency of recurrence and time until recurrence were similar in treated children and controls [63].

In vitro studies have also shown significant activity of silver nanoparticles against biofilm-producing otopathogens [64]. However, despite these and several other antibiofilm measures being developed and tested in vitro, none have been evaluated in double-blind, randomized, placebo-controlled clinical trials. Therefore, traditional prophylactic and therapeutic approaches to rAOM and csOM, including antibiotic prophylaxis or long-term antibiotic administration, respectively, remain the only currently available solutions for managing these diseases.

### 5.2. Otitis Media with Effusion

Otitis media with effusion (OME) is a common clinical condition characterized by the presence of fluid in the middle ear space without signs of acute infection. About 80% of preschool children experience at least one episode of OME, 30–40% have recurrent episodes, and 5–10% have chronic disease lasting for more than a year [65,66]. While most cases resolve spontaneously within a few weeks without significant clinical problems, recurrent and persistent cases can lead to long-term hearing loss, resulting in significant behavioral and educational difficulties [65]. OME generally develops following an upper respiratory tract infection or an AOM episode, with more severe cases closely linked to frequent recurrences of these diseases.

For years, OME was not considered an infectious disease but rather a condition resulting from eustachian tube dysfunction and poor middle ear ventilation following upper respiratory tract infections. This belief was based on traditional culture methods, which found positive results in only a minority of children with OME [67], and the generally poor effectiveness of antibiotic treatment on the disease’s clinical course [68]. However, recent studies using molecular biology methods or alternative microbiological approaches have significantly altered our understanding of OME. Evidence that bacterial DNA can be detected in more than 80% of effusions in OME patients, compared to 32% positivity with traditional cultures, strongly supports the idea that OME may primarily be due to bacteria [69,70]. Similar conclusions were drawn when bacterial viability stains and confocal laser scanning microscopy were used to evaluate fluid characteristics, with live bacteria demonstrated in 91.8% of samples [71].

The discovery of otopathogen-derived biofilms in the middle ear mucosa of OME patients suggests that biofilm formation may play a significant role in long-term OME development and explains why antibiotic treatment is generally ineffective. The poorly ventilated middle ear provides an excellent environment for bacterial mucosal biofilm formation. *Sp*, nt*Hi*, and *Mc*, alone or in various combinations, are the most commonly detected pathogens. However, other pathogens such as Ps, Sa, *Diphtheroids*, *Streptococcus viridans*, and *Neisseria* are more frequently found in OME than in rAOM or csOM [67]. Interestingly, the degree of hearing loss in children with OME was significantly higher in patients with detected bacterial biofilms [67].

### 5.3. Recurrent Adenoiditis

Upper respiratory tract infections caused by viruses are extremely common in the first years of life, mainly due to the immaturity of the immune system [72]. Moreover, the upper respiratory mucosa is colonized by numerous bacteria, including those that are the main causes of acute and chronic bacterial rhinosinusitis and middle ear disease. The adenoidal pads, located at the dome of the nasopharynx, are particularly prone to repeated infections with biofilm-producing bacteria due to their anatomical position. The enlargement of lymphoid tissue during childhood further contributes to this problem [73]. Practically, adenoids can be chronically infected and simultaneously act as a reservoir of biofilms, periodically spreading biofilm-related pathogens to the middle ear, leading to recurrent and chronic middle ear and sinus diseases such as AOM. Bacterial biofilm can be found in up to 100% of nasopharyngeal samples taken from children with chronic adenoiditis, although the reported prevalence rate varies widely (41–100%) depending on the sampling and microbiological analysis techniques used [74].

In addition to chronic adenoiditis itself, biofilm location appears to be a critical factor associated with the risk of rAOM. This is strongly supported by a study involving children with chronic adenoiditis associated with recurrent AOM. Differences in the prevalence, strains, and biofilm production of pathogens collected from the nasopharyngeal dome and the nasopharyngeal ostium of the eustachian tube were evaluated. Biofilm-producing bacteria, mainly *Sp* and *Mc*, were more frequently located near the ostium of the eustachian tube than at the nasopharyngeal dome (72.2% vs. 53.3%; *p* = 0.04) [75].

### 5.4. Protracted Bacterial Bronchitis and Non-Cystic Fibrosis Bronchiectasis 

Non-cystic fibrosis (non-CF) bronchiectasis is a relatively common complication in children with recurrent episodes of protracted bacterial bronchitis (PBB) [76,77]. These conditions can significantly impact child health, as they are associated with chronic respiratory insufficiency and early death. Development is strongly linked to repeated lower respiratory tract infections, resulting in persistent endobronchial inflammation and progressive tissue damage. Unfortunately, definitive data on the potential beneficial role of anti-biofilm therapy in children with these clinical manifestations are lacking.

Recently, a significant advance in understanding the importance of biofilms in causing PBB and non-CF bronchiectasis was made through a well-conducted study by Marsh et al. In a cross-sectional study, they evaluated the presence and prevalence of biofilms in bronchoalveolar lavage (BAL) specimens from 69 children with PBB and 75 children with non-CF bronchiectasis [78]. The study also examined the association of biofilms with signs of lower respiratory tract infection (LRTI) and determined whether the biofilm originated from the upper or lower airways. LRTI was diagnosed when the neutrophil percentage in BAL was ≥15% or when BAL culture was positive for a bacterial pathogen in concentrations ≥10^4^ colony-forming units per mL. Biofilms were considered to be derived from the upper respiratory tract when high concentrations of squamous epithelial cells were detected in the BAL, as these cells are not typically found in the lower airways.

The results indicated that biofilms were prevalent, though not ubiquitous, in the BAL of children with PBB or bronchiectasis. Lower airway biofilms were detected in 36% and 41% of children with PBB and bronchiectasis, respectively. Additionally, these biofilms were more common in patients with LRTIs than in those without (48% vs. 19%; OR 5.28; 95% CI 2.20–13.83), with non-typeable nt*Hi* and *Mc* being the most common infectious pathogens. These findings strongly suggest a role for biofilms in the development of non-CF-related chronic infectious diseases of the respiratory tract and indicate that anti-biofilm therapy could benefit at least some children with these conditions. This conclusion is particularly important considering that BAL may underestimate the true prevalence of biofilms in the respiratory tract [79].

Regarding upper airway biofilms, they were more common among children with bronchiectasis (43%) than in children with PBB (23%). Similarly, upper airway biofilms and LRTI were more frequent in children with bronchiectasis (19%) than in those with PBB (6%). The role of upper airway biofilms in PBB and bronchiectasis remains unclear. Their presence in BAL could be considered a result of contamination rather than a contributing factor in the development of PBB and bronchiectasis. However, since the prevalence of squamous epithelial cells in BAL was generally low, it is hypothesized that, in some cases, upper airway biofilms were microaspirated from the nose and pharynx and could play a pathogenic role in chronic infections of the lower respiratory tract. In patients with bronchiectasis, chronic cough and poor respiratory clearance can favor microaspiration [80].

Despite these advances, further studies are needed to clarify the role of biofilms in PBB and bronchiectasis development and to identify which children might benefit from anti-biofilm treatment. However, the findings of this study underscore the importance of including biofilm testing and subtyping in children who suffer from recurrent protracted bacterial bronchitis or non-CF bronchiectasis with recurrent infectious episodes.

### 5.5. Cystic Fibrosis

Recurrent bacterial airway infections, mainly due to *Pa* and *Sa*, are recognized as the most common and severe clinical manifestations of cystic fibrosis (CF) [81]. These infections are the primary cause of progressive lung function decline leading to the end-stage lung disease frequently seen in CF patients [82]. While *Pa* predominates in adults, *Sa* is the most common opportunistic pathogen in children and adolescents, with 60% to 80% of CF patients under 20 years old being colonized [83].

CF is an autosomal recessive disorder resulting from mutations in the CF transmembrane conductance regulator gene, which controls mucus viscosity on epithelial surfaces. Increased mucus viscosity impairs mucociliary clearance and favors bacterial growth and biofilm formation. The association of biofilm formation with recurrent, difficult-to-treat infections and poor prognosis in CF has been repeatedly demonstrated in both adults and children [84]. Interestingly, the increased expression of genes regulating biofilm development is associated with a higher risk of antibiotic eradication failure in chronic *Pa* and *Sa* infections [85].

The direct relationship between biofilm detection and the negative clinical evolution of lung function in CF patients has led researchers to evaluate whether measures found effective in vitro or in experimental animals for disrupting biofilms could also be effective in humans. Most studies have focused on adults with chronic *Pa* infections.

A randomized, double-blind, placebo-controlled, crossover study enrolled 65 CF adults with chronic *Pa* infection to test OligoG, a low molecular weight oligomer from alginate. Although OligoG was safe, the primary endpoint—an absolute change in the percentage predicted forced expiratory volume in 1 s (FEV1) at the end of the 28-day treatment—was not met. Only a nonsignificant improvement in FEV1 was observed [86]. Despite this result, further studies were planned (NCT03698448 and CT03822455), but the first was withdrawn, and the status of the second is currently unknown.

An initial pilot randomized clinical trial involving 12 CF adults with chronic *Pa* infections found that adding 10 ppm inhaled nitric oxide (NO) inhalation (8 h/day for 5–7 days) to antibiotic therapy significantly reduced *Pa* biofilm aggregates compared with a placebo [87]. A subsequent trial to administer a higher dose (160 ppm) of inhaled NO in CF patients was planned (NCT02498535) but was terminated due to safety concerns related to COVID-19. Additionally, substances that could increase endogenous NO production were considered. A Phase 1B randomized, double-blind, placebo-controlled trial investigated CB-280, a potent arginase inhibitor (NCT04279769), but results are not yet available [88].

A meta-analysis of four trials involving CF patients showed that those treated with azithromycin had improved lung function, especially in the subgroup colonized with *Pa* [89]. There are other hypotheses regarding the clinical efficacy of azithromycin, including its effects on airway inflammation [90]. Additionally, ceftazidime and ciprofloxacin have been shown to disrupt quorum sensing (QS), suggesting that targeting biofilm bacteria using this mechanism holds promise [91].

A recent Phase 2 study conducted in 23 centers in the U.S. demonstrated trends toward improved lung function and lower *Pa* density in sputum in CF patients with chronic Pa infection treated with intravenous (IV) gallium nitrate [92]. An ongoing study is evaluating IV gallium for treating nontuberculous mycobacterial infections in CF patients (NCT04294043).

Several case reports, including some involving children, have shown that phage administration by nebulizer or the intravenous route to CF patients with chronic *Pa* and *Sa* infections has resulted in decreased sputum bacterial densities, improved patient health, and extended time until subsequent bacterial infection [93,94]. However, a U.S. study found that while phage nebulization once a day for 7 days was well tolerated and safe, it did not significantly benefit patients. No difference in sputum *Pa* content or lung function was observed between treated patients and controls [95]. A Phase 1b/2 study of a single dose of an intravenous mixture of bacteriophages targeting *Pa* in adult CF patients is currently recruiting participants (NCT05453578).

Despite these advances, further research is needed to fully understand the role of biofilms in CF and to identify which patients might benefit from antibiofilm therapies. The findings of these studies highlight the importance of including biofilm testing and subtyping in managing CF patients with recurrent infections.

## 6. Conclusions

Respiratory infections such as AOM, OME, adenoiditis, and PBB are extremely common in children. Moreover, infectious lower respiratory tract diseases are a major cause of progressive lung function deterioration in patients with CF. Recurrent and chronic cases are frequently associated with biofilm-producing pathogens on the respiratory mucosa. Studies have elucidated the mechanisms of biofilm development and explained why biofilms can lead to antibiotic treatment failure and the development of recurrent and chronic diseases. However, the clinical relevance of biofilm in the different diseases and in the different subjects should be better defined. Further studies in this regard are needed to identify which subjects can benefit from the recent developed therapeutic measures. Several antibiofilm measures have been developed, and in some cases, in vitro and animal studies have shown significant effectiveness, suggesting potential use in humans. However, their clinical use has been limited, especially in children. Traditional approaches to treating rAOM, csOM, OME, adenoiditis, and infections secondary to bronchiectasis and CF have remained largely unchanged, even when biofilm-forming pathogens had been detected. Translating findings from in vitro and experimental studies to clinical practice in humans is challenging and requires extensive knowledge and clinical trials that are not yet available. Additionally, addressing the issue of biofilms in pediatrics cannot be fully resolved until a protocol that ensures rapid biofilm identification is established. However, as a transitional measure while waiting for drugs specifically effective in biofilm destruction, a proper use of antibiotics should be considered. As some of them, such as azithromycin, have been found effective in reducing biofilm, their use at the highest dosage should be preferred, at least in those patients in whom the infectious pathogen is sensitive to these drugs. It is much more difficult to give potentially useful suggestions for the treatment of children with CF, especially those with biofilm-associated frequent recurrences and rapid lung function deterioration. For these subjects, one can only hope that some of the few measures that are supposed to have a certain effectiveness can confirm their activity in the pediatric population and become quickly available for use in children. However, the idea of eradicating the biofilm with prolonged antibiotic treatments is to be totally excluded. Long-term antibiotic administration appears to be associated with more risks (including an increase in antimicrobial resistance and antibiotic-related adverse effects) than benefits and cannot be considered the optimal approach in routine practice.

## Figures and Tables

**Table 1 microorganisms-12-01522-t001:** Biofilm-producing pathogens in pediatric respiratory infection.

Pathogen	Common Infections	Mechanism of Biofilm Formation	Antibiotic Resistance Mechanism
*Streptococcus pneumoniae* (*Sp*)	Acute Otitis Media (AOM), Sinusitis	DNA adenine methyltransferase, quorum sensing molecules	Poor penetration of antibiotics, persister cells
*Non-typable Haemophilus influenzae* (*ntHi*)	Otitis Media, Sinusitis, Pneumonia	Type IV pilus, DNABII family of proteins	Enzyme inactivation of antibiotics, persister cells
*Pseudomonas aeruginosa* (*Pa*)	Cystic Fibrosis, Bronchiectasis	Alginate, pel and psl polysaccharides, eDNA	Efflux pumps, enzyme degradation, biofilm matrix barrier
*Staphylococcus aureus* (*Sa*)	Cystic Fibrosis, Skin Infections	Polysaccharide intercellular adhesin (PIA), ica operon	Altered target sites, biofilm matrix barrier
*Moraxella catarrhalis* (*Mc*)	Otitis Media, Sinusitis	Adhesins, biofilm matrix components	Efflux pumps, biofilm matrix barrier
*Streptococcus pyogenes* (*Spy*)	Pharyngitis, Otitis Media	Adhesins, extracellular matrix proteins	Efflux pumps, altered target sites

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
