# Peer review of "Biofilm Production and Its Implications in Pediatrics"

_microorganisms, 2024, doi:10.3390/microorganisms12081522_

Round 1
Reviewer 1 Report
Comments and Suggestions for Authors
The manuscript thoroughly reviews the topic of biofilm production in pediatric respiratory infections, covering various aspects from pathogen identification to potential treatment strategies. The manuscript is well-organized; however, it is too narrative and lacks original data as well as a discussion on the limitations of clinical trials. Please revise your manuscript to address the following points:
#1: To provide a clearer picture of the current state of clinical research in this area, a more detailed discussion of existing clinical trials, their methodologies, and outcomes should be included.
#2:If possible, incorporate original data or case studies to strengthen the manuscript's contribution to the field.
#3: Provide a more critical assessment of the studies reviewed, discussing potential biases, limitations, and areas for future research.
#4: Incorporate more figures and tables to summarize key points and data, which can help in visualizing complex information.
Comments on the Quality of English Language
English is not so bad, but this need to be checked in detail from grammatical viewpoint.
Author Response
The manuscript thoroughly reviews the topic of biofilm production in pediatric respiratory infections, covering various aspects from pathogen identification to potential treatment strategies. The manuscript is well-organized; however, it is too narrative and lacks original data as well as a discussion on the limitations of clinical trials.
Re: Thank you for your comments. We revised the manuscript according to your recommendations.
Please revise your manuscript to address the following points:
#1: To provide a clearer picture of the current state of clinical research in this area, a more detailed discussion of existing clinical trials, their methodologies, and outcomes should be included.
Re: A detailed discussion of all the existing clinical trials is reported in the manuscript.
#2:If possible, incorporate original data or case studies to strengthen the manuscript's contribution to the field.
Re: All the published studies on the topic are presented.
#3: Provide a more critical assessment of the studies reviewed, discussing potential biases, limitations, and areas for future research.
Re: The text of the Conclusion section has been largely improved.
#4: Incorporate more figures and tables to summarize key points and data, which can help in visualizing complex information.
Re: There are very few studies in humans on the topic. For this reason we think that it is better to maintain only one Table and then leave the possibility for the reader to evaluate the text that is very balanced.
Reviewer 2 Report
Comments and Suggestions for Authors
The article deals with basic concepts of the creation and maturation of biofilm by different bacteria involved in respiratory diseases. Although it is not the field of my dental specialty, the concepts and aspects handled by the authors are correct and well applied to the biology of the bacterial biofilm. According to the biography handled, the narrative review is coherent and interesting and gives a current vision of how to manage these respiratory diseases linked to biofilm. In my view it is an interesting narrative review that is easy to read and well documented. However, I consider that my opinion should be considered second only to that of respiratory specialists.
Author Response
The article deals with basic concepts of the creation and maturation of biofilm by different bacteria involved in respiratory diseases. Although it is not the field of my dental specialty, the concepts and aspects handled by the authors are correct and well applied to the biology of the bacterial biofilm. According to the biography handled, the narrative review is coherent and interesting and gives a current vision of how to manage these respiratory diseases linked to biofilm. In my view it is an interesting narrative review that is easy to read and well documented. However, I consider that my opinion should be considered second only to that of respiratory specialists.
Re: Thank you very much for the appreciation of our manuscript. We improved the text according to the suggestions received from the other two reviewers.
Reviewer 3 Report
Comments and Suggestions for Authors
The article “Biofilm Production and Its Implication in Pediatrics” is devoted to a current topic.
It is known that biofilms reduce antibiotic efficacy, contributing to treatment failure and disease persistence. This narrative review discusses biofilm production by respiratory pathogens such as Streptococcus pneumoniae, Haemophilus influenzae, Pseudomonas aeruginosa, and Staphylococcus aureus. It examines their mechanisms of biofilm formation, antibiotic resistance, and the challenges they present in clinical treatment. However, the clinical application of many antibiofilm strategies, particularly in children, remains limited.
The text of the article is well structured.
Minor notes:
1. In section 2 “Biofilm Formation by Respiratory Bacteria” the authors write in detail about the formation of biofilms, the matrix of which is based on polysaccharides. This section should include material about biofilms, the matrix of which is dominated by protein components.
2. In section 6 “Conclusion”, more material should be added about the prospects for treatment of biofilms in pediatrics, as well as about new tendencies and trends in this area.
Technical shortcomings:
1. Throughout the text of the article, the latin names Streptococcus pneumoniae, Haemophilus influenzae, Pseudomonas aeruginosa, Staphylococcus aureus (and not only) should be italicized, including in Table 1.
2. Throughout the text of the article, in vitro and in vivo should be italicized.
3. The design of the references is very strange; this should be corrected.
Author Response
The article “Biofilm Production and Its Implication in Pediatrics” is devoted to a current topic.
It is known that biofilms reduce antibiotic efficacy, contributing to treatment failure and disease persistence. This narrative review discusses biofilm production by respiratory pathogens such as Streptococcus pneumoniae, Haemophilus influenzae, Pseudomonas aeruginosa, and Staphylococcus aureus. It examines their mechanisms of biofilm formation, antibiotic resistance, and the challenges they present in clinical treatment. However, the clinical application of many antibiofilm strategies, particularly in children, remains limited.
The text of the article is well structured.
Re: Thank you very much for your positive evaluation. We improved the text according to your suggestions.
Minor notes:
- In section 2 “Biofilm Formation by Respiratory Bacteria” the authors write in detail about the formation of biofilms, the matrix of which is based on polysaccharides. This section should include material about biofilms, the matrix of which is dominated by protein components.
Re: We revised the text according to your suggestions (p. 7).
- In section 6 “Conclusion”, more material should be added about the prospects for treatment of biofilms in pediatrics, as well as about new tendencies and trends in this area.
Re: The text has been improved as recommended (p. 12).
Technical shortcomings:
- Throughout the text of the article, the latin names Streptococcus pneumoniae, Haemophilus influenzae, Pseudomonas aeruginosa, Staphylococcus aureus (and not only) should be italicized, including in Table 1.
Re: Done as recommended.
- Throughout the text of the article, in vitro and in vivo should be italicized.
Re: Done as suggested.
- The design of the references is very strange; this should be corrected.
Re: We used the format suggested by Instruction for authors.